# The Role of Cluster C19MC in Pre-Eclampsia Development

**DOI:** 10.3390/ijms232213836

**Published:** 2022-11-10

**Authors:** Ilona Jaszczuk, Izabela Winkler, Dorota Koczkodaj, Maciej Skrzypczak, Agata Filip

**Affiliations:** 1Department of Cancer Genetics with Cytogenetic Laboratory, Medical University of Lublin, Radziwillowska Street 11, 20-080 Lublin, Poland; 2Second Department of Gynecological Oncology, St. John’s Center of Oncology of the Lublin Region, Jaczewski Street 7, 20-090 Lublin, Poland; 3Second Department of Gynecology, Lublin Medical University, Jaczewski Street 8, 20-954 Lublin, Poland

**Keywords:** pregnancy, pre-eclampsia, microRNAs, C19MC

## Abstract

Pre-eclampsia is a placenta-related complication occurring in 2–10% of all pregnancies. miRNAs are a group of non-coding RNAs regulating gene expression. There is evidence that C19MC miRNAs are involved in the development of the placenta. Deregulation of chromosome 19 microRNA cluster (C19MC) miRNAs expression leads to impaired cell differentiation, abnormal trophoblast invasion and pathological angiogenesis, which can lead to the development of pre-eclampsia. Information was obtained through a review of articles available in PubMed Medline. Articles on the role of the C19MC miRNA in the development of pre-eclampsia published in 2009–2022 were analyzed. This review article summarizes the current data on the role of the C19MC miRNA in the development of pre-eclampsia. They indicate a significant increase in the expression of most C19MC miRNAs in placental tissue and a high level of circulating fractions in serum and plasma, both in the first and/or third trimester in women with PE. Only for miR-525-5p, low levels of plasma expression were noted in the first trimester, and in the placenta in the third trimester. The search for molecular factors indicating the development of pre-eclampsia before the onset of clinical symptoms seems to be a promising diagnostic route. Identifying women at risk of developing pre-eclampsia at the pre-symptomatic stage would avoid serious complications in both mothers and fetuses. We believe that miRNAs belonging to cluster C19MC could be promising biomarkers of pre-eclampsia development.

## 1. Introduction

In the course of each pregnancy, the placenta undergoes an extremely dynamic and necessary remodeling, as a result of which free nucleic acids, including miRNAs, are released into the blood of pregnant women [1]. Over the last 20 years, the increasing knowledge of the processes taking place in the placenta and of the physiological and pathological importance of miRNAs allow for attempts to use them as non-invasive biomarkers. MicroRNAs (miRNAs) are a class of non-coding, single-stranded RNA (ncRNA) that are 19–25 nucleotides long [2]. The on-line-available and continuously updated version 22.1 of the miRNA miRBase database contains a list of 1917 annotated hairpin precursors and 2654 mature sequences contained in the human genome [3].

miRNAs are involved in post-transcription regulation of the expression of more than 60% of all human genes through a process called “silencing”, by inhibiting translation of messenger RNA (mRNA) or mRNA cleavage processes [4]. The structurally mature miRNAs are functionally part of the RNA-induced silencing complex (RISC), which also contains proteins from the Argonaute family (Ago2). The interaction between miRNAs and the mRNA target sequence takes place de facto between the complementary 7-nucleotide “seed” fragment in the miRNAs and the 3’ untranslated regions (3’UTR) of the mRNA [5,6,7,8]. Possible interactions between miRNAs and other gene regions, including 5’ UTR (the coding sequence and promoter), have also been reported [9]. The regulation of the expression of single gene can come about due to the influence of a variety of miRNAs, while, at the same time, certain miRNA can regulate the expression of different genes [10,11]. It is likely that miRNAs form specific regulatory networks that cooperate in the repression of target genes.

miRNAs are divided into tissue-specific or circulating. Surprisingly, circulating miRNAs are highly stable in body fluids, especially serum, because they appear to be resistant to endogenous ribonuclease activity and may be freely circulating or associated with extracellular structures such as exosomes [12]. Circulating miRNAs can be collected for analysis in a minimally invasive manner and it makes them ideal biological markers (biomarkers). This is of particular importance in the context of early detection of the risk of developing placenta-related pregnancy complications, such as gestational hypertension (GH), pre-eclampsia (PE) or intrauterine growth restriction (IUGR).

According to studies on the pathogenesis of pregnancy complications related to the presence of trophoblast (PIRC—placental complications associated with placental insufficiency) [13], microRNAs can be divided into four groups: placental-associated; placental-specific; placenta-derived circulating and uterus microRNA. Detection of C19MC miRNAs expression in placental tissue and in maternal circulation confirms the involvement of C19MC in the physiological and abnormal development of the placenta (Figure 1) [13,14,15].

A fraction of specific circulating microRNAs derived from placenta can be obtained from whole blood, as well as in the blood serum of pregnant women [13]. MicroRNAs associated with the presence of trophoblasts are detected in the serum during pregnancy and disappear after delivery [14]. Interestingly, Cretoiu et al. indicated the possibility of qualitative differences in the microRNA profile from representatives of different populations [13]. Weber et al. provided the possibility of collecting body fluids in a non-invasive way (mother’s milk, colostrum, saliva, seminal fluid, tears and urine) and with invasive methods (amniotic fluid, cerebrospinal fluid, plasma, bronchial lavage, pleural fluid and peritoneal fluid), also indicated that the concentrations and compositions of the miRNAs measured in them vary [15].

According to the literature and WHO recommendations, pre-eclampsia develops as a serious complication in 2–10% of all pregnancies, increasing maternal and perinatal mortality and mortality. Pre-eclampsia is now defined as arterial hypertension developing de novo after 20 weeks of age of pregnancy coexisting with: newly diagnosed proteinuria and/or damage to the mother’s kidneys; maternal liver dysfunction; neurological symptoms; or hemolysis thrombocytopenia and/or IUGR [16,17].

The causes of pre-eclampsia include defective trophoblast implantation, abnormal uteroplacental perfusion and ischemia, abnormalities in the maternal immune response and systemic inflammatory response. In the early stages of pregnancy, an undoubted factor for the development of pre-eclampsia is incorrect implantation, caused by disturbances in division, migration and invasion of extravillous trophoblasts (EVT) [18,19,20]. Factors involved in the regulation of the trophoblast implantation process are adhesion molecules, growth factor receptors and ligands (EGF, TGFβ, IGF), integrins, enzymes responsible for the breakdown of the extracellular matrix and angiogenesis [21]. The developing trophoblast is a source of EVT involved in the reconstruction of the maternal spiral arteries: destroying smooth muscles and elastin in the walls of spiral arteries and their replacement through fibrinoid material [20]. This causes changes of key importance for the development of placental circulation: reducing the speed and pulsation of the impact of maternal blood and protecting the delicate villi and microvilli of the placenta against damage. The histopathological examination of pre-eclampsia placentas revealed: abnormal remodeling of maternal spiral arteries; acute atherosclerosis with fibrin necrosis; and accumulation of lipid-laden intimal macrophages. In addition, the syncytiotrophoblast hyperplasia with degeneration or apoptosis can lead to the release of trophoblast debris, cell-free DNA, exosomes, pro-inflammatory factors and anti-angiogenic factors for the circulation of a pregnant woman [22]. Most of the theories in the literature emphasize the importance of the vascular factor that is related to the placental vasoconstriction or damage to the endovascular endothelium [18,19]. Among the potential molecular mediators of the development of pre-eclampsia, soluble vascular receptor endothelial growth factor (sFLT) (VEGF) and placental increase (PlGF) are listed as significant. Increasing the sFLT level binds VEGF, limiting its bioavailability to the maternal vascular endothelium. In turn, the reduction in endogenous nitric oxide production leads to vasoconstriction [23]. The sequelae of pre-eclampsia development may be short-term or long-term, including long-term cardiac complications in women and their offspring [24,25].

Growing evidence suggests that non-coding RNAs (ncRNAs) are involved in the development of pre-eclampsia. Many ncRNAs, including microRNAs and long non-coding transcripts, show rich expression in the placenta and a specific profile of placental complications. Buckberry et al. highlighted, in a review of studies using genome scale and single gene expression, the importance of epigenetically regulated ncRNAs through genomic imprinting, including the C19MC miRNAs, and confirmed their importance in the development of PE. An important question, however, is whether the change in miRNA expression levels is a cause or a consequence of PE [26].

## 2. The Chromosome 19 MicroRNA Cluster (C19MC)

The chromosome 19 microRNA cluster (C19MC) is a primate specific miRNA cluster located on the human chromosome 19q13.41 that is 100 kb in length. It contains 46 tandem repeating microRNA genes encoding 58 mature miRNAs [27,28]. C19MC is found only in primates and is almost exclusively expressed in the placenta, although low levels have also been shown in embryonic stem cells, testes and some tumors [29,30,31,32]. Many publications emphasize that the synthesis of miRNAs is highly orchestrated, and that in the placenta they are expressed at the appropriate time, in a tissue and species manner [33,34,35].

The level of C19MC miRNA expression in plasma and placenta increases with the advancing age of pregnancy, decreasing sharply after delivery [36,37]. In the plasma of pregnant women, C19MC miRNAs form a part of the placental-related fraction of circulating miRNAs or are packed into exosomes. The source of C19MC miRNAs in the plasma of pregnant women are cells of various areas of the placenta in which cells express C19MC (which can be studied after delivery), and also cells that during remodeling, undergo apoptosis, releasing placental debris and secreting exosomes into the maternal circulation [38,39].

Exosomes or nanovesicles are a small fraction (30–150 nm) of the extracellular vesicles (EVs) formed in multivesicular bodies (MVBs) [40] that are released by most cells into the extracellular space. Their function is to mediate intercellular communication through signaling molecules packed inside and secreted during exocytosis (proteins, lipids, RNA and DNA) after fusion with the cell membrane of target cells. Increased oxidative stress, observed in pre-eclampsia development, predisposes syncytiotrophoblast (STB) cells to the production of more humoral factors and to the release of microvesicles, including exosomes [41,42]. The composition of the miRNAs transported in exosomes in the PE is different compared to normal pregnancies [43]. Extracellular miRNAs packed into exosomes can be responsible for intercellular communication in an autocrine or paracrine manner, and at greater distances through circulation [44,45]. They can also modulate the immune response that ensures immune tolerance in the mother–fetus relationship or modify pro-inflammatory reactions in the course of pregnancy [46,47].

C19MC belongs to the genes encoding miRNAs whose expression is influenced by genomic imprinting, an epigenetic mechanism related to monoallelic expression in a parent-of-origin manner. C19MC is expressed exclusively from the paternal allele in the placenta which has been confirmed by single nucleotide polymorphism genotyping (SNP: G or T, rs55765443) mapping upstream the most 5′ microRNA transcribed by C19MC [48]. Many C19MC miRNAs are likely formed from the introns of a poorly characterized transcript called ”C19MC-HG”, composed of many repeating non-coding exons [27]. C19MC is expressed by the polymerase II promoter region, approximately 17 kb from the first exon, overlapping the differential methylation (DMR) region. The promoter is rich in CpG, showing the maternal characteristic methylation imprint acquired in the oocytes [48]. Maternal-specific methylation is deposited at CpG1, here termed ”C19MC-DMR1” (C19MC- differentially methylated region 1).

The structure and function of the C19MC cluster corresponds to features of imprinted genes present in the human genome. Imprinted genes form large chromosomal domains (up to 3 Mb), most of which are expressed in the placenta [49,50] and play an important role in prenatal embryo or placenta growth or regulate metabolic pathways in the placenta [51,52,53]. Regulation of imprinting gene expression in a given cluster is coordinated by epigenetically modified imprinting control regions (ICRs) that acquire a parental specific male DNA methylation imprint or female germline. In addition, a convergence has been observed between ICRs with maternal imprint and CpG-rich promoter regions [54]. DNA methylation at ICRs of imprinted genes is acquired during gametogenesis. Although methylation is a reversible process, the pattern of ICRs’ methylation is refractory to the genome-wide methylation reprogramming that occurs in the embryo after fertilization. DNA methylation levels can also be modified by the presence of specific SNPs [55], adjacent to the CpG islands in the in-cis system. It has been shown that C19MC miRNA transcription can be activated in cells by the use of DNA methylation inhibitors, confirming methylation-dependent epigenetic control in this region [48]. ICRs are themselves also marked by allele-specific post-translational histones modifications [28].

## 3. Results

The most important data from the analyzed papers based on the following keywords: “C19MC and pre-eclampsia” and “C19MC and preeclampsia” are presented in Table 1 [26,28,35,38,39,43,56,57,58,59,60,61,62,63,64].

## 4. Materials and Methods

Data indicating the diagnostic value of C19MC miRNAs in predicting pre-eclampsia were analyzed from publications available in PubMed (https://pubmed.ncbi.nlm.nih.gov/ accessed on 12 September 2022) [65]. A search of the PubMed database was firstly conducted in September 2022. The scope of the search was determined based on the following keywords: ”C19MC and pre-eclampsia”, obtaining the result of 16 articles, and “C19MC and preeclampsia”, obtaining the result of 11 articles. Of note, 11 articles published in 2013–2022 were common for both search scopes. The article by Biro et al. (2018) written in Hungarian was not analyzed (Table 1) [26,28,35,38,39,43,56,57,58,59,60,61,62,63,64].

Then, in November 2022, another analysis of publications available in the PubMed database was carried out, extending the scope of the search by the following keywords: “MiR-515/miR-516/miR-517/miR-518/miR-519/miR-520/miR-525/miR-526” and “preeclampsia”. A total list of 30 articles was obtained. Some articles were duplicated for different miRNAs. Taking this into account, 17 articles were finally analyzed. A total of 4 articles were excluded: 1 written in Czech (Hromadnikova et al., 2010); 2 experimental works on cell lines (Logan et al., 2022; Shi et al., 2022); 1 review (Sheikh et al., 2016). The other 10 articles listed in Table 2 were included based on a systemic review by Ogoyama et al., (2022) [34,38,39,56,58,60,63,66,67,68,69,70,71,72,73,74,75,76,77,78,79,80].

## 5. Discussion

Chim et al., in 2008, were the first to highlight the potential use of miRNAs as biomarkers of pregnancy complications [81,82], while the chromosome 19 microRNA cluster (C19MC) was first described by Bentwich et al., in 2005 [83]. Noguer-Dance et al., in turn, showed that C19MC microRNAs were clearly expressed during embryo and placenta development. Moreover, they recognized that the imprinted C19MC miRNA genes had to evolve to improve the signaling pathways underlying primate morphology and placental development [43,48]. Their work also revealed that C19MC dysregulation leads to dysfunctional trophoblast cells, abnormal placentation and the consequent development of PE [43]. Zhang et al., in a subsequent study, showed that miR-515-5p was significantly decreased during the differentiation of human syncytiotrophoblasts and significantly increased in the placenta during the development of pre-eclampsia. In contrast, miR-515-5p overexpression inhibited the differentiation of the syncytiotrophoblast [60].

Inno et al., in their 2021 research, assessed the expression profile of the human miRNome and the dynamics of its changes in the placenta of pregnant women in three trimesters, and looked for relationships with the occurrence of pregnancy complications. Among the obtained conclusions, they pointed out that most of the C19MC miRNA target genes were involved in cell signaling or transcription regulation important in early pregnancy. Accordingly, two thirds of the C19MC miRNA is expressed especially in the first trimester, is very low in the second trimester and slowly increases in the third trimester [55]. Additional research suggested that the advantage of expression primate, paternal-specific C19MC in the first trimester was likely to be associated with a dose-dependent effect on placental transcripts [8,63].

Hromadnikova et al., on the basis of their conducted research, held that C19MC miRNAs were expressed only in placental tissues (miR-520a, miR-516-5p, miR-517, miR-518b, miR-519a, miR-524-5p, miR-525, miR- 526a, miR-526b, miR-520h) [1,56,84]. They also demonstrated the importance of C19MC in the development of placenta-related complications (pre-eclampsia) and pregnancy hypertension or fetal growth restriction (IUGR) [56]. In subsequent studies, Hromadnokova et al. indicated a strong correlation between the increased expression in the first trimester of miR-516-5p, miR-517, and especially miR-520h and miR-518b, and the risk of gestational hypertension [85]. The positive correlation between the increase in expression in the first trimester (12–14 hbd) of miR-520a in the serum of pregnant women who developed severe pre-eclampsia was also previously reported by Ura et al. [71]. Further studies by Hromadnikova et al. confirmed the increased expression of miR-517-5p, miR-518b and miR-520h in the serum of pregnant women tested in the first trimester (11–13 hbd) who developed pre-eclampsia. Herein, miR-517-5p had the highest predictive value. Unfortunately, no correlation was found between the level of C19MC expression and the risk of IUGR [38].

Miura et al. assessed the level of expression miR-520a-5p, miR-520h, miR-516a-5p, miR-516b, miR-518b, miR-519d, miR-525-5p, miR-515-5p, miR-526b, miR-1323 in the plasma of pregnant women at 27–34 weeks of gestation. They observed upregulation of expression of C19MC miRNAs in pregnant women with severe pre-eclampsia (sPE) [86]. Jiang et al., in turn, noticed in a patient with sPE, an increased concentration of miR-520g in the serum already in the first trimester [74]. In two studies from 2014 and 2015, an inverse correlation between the expression level of miRNAs on C19MC and the weight of the placenta and birth weight of newborns was found [59,86]. Other researchers have reported that the level of C19MC expression increases with the advancement of pregnancy [1,66,84] and is higher in the case of early onset PE ((PEEO); <34 weeks of gestation) than late onset PE ((PELO); >37 weeks of gestation) [59].

Chaiwangyen et al. indicated the importance of miR-519d-3p in the formation of immune tolerance in pregnancy by influencing the proliferation and migration of maternal immune cells (monocytes, granulocytes, T-lymphocytes and NK cells) [87]. However, it should be emphasized that humoral factors and miRNAs contained in exosomes also affect the maternal vascular endothelium, stimulating it to release cytokines and activating neutrophil adhesion, and as a result, inducing a systemic inflammatory response characteristic of advanced PE [43]. In addition, Delorme Axford et al. clearly indicated that C19MC miRNAs (miR517-3p, miR-512-3p, or 516b-5p) can increase the resistance of maternal cells at the fetal–mother interface to viral infection, by induction of autophagy [88,89]. Moreover, miR-517a-3p was found to influence the activation of maternal T-lymphocyte and NK cell proliferation, and via the PRKG1 gene, the on activation of the nitric oxide/cGMP signaling pathway [89].

Zhao Z. et al., in their review on the use of miRNAs as potential biomarkers for assessing the risk of pregnancy complications, pointed to conflicting data on the expression of individual miRNAs in the various cited studies. As reasons for this, they cited the possible impact of the following heterogeneity factors in patient populations: ethnic origin; variability in the severity of PE; variability of the gestational age; maternal interview; route of delivery or other test conditions: origin and processing samples (tissues, cells, serum and plasma); and data analysis [82]. Furthermore, they pointed out that the use of C19MC as a biomarker of pre-eclampsia development is somewhat complicated by the fact that some miRNAs: miR-16; let-7d; miR-520a *; miR-520h; miR-525; miR-516-5p; miR-517 *; and miR-518b are not stable enough during long-term frozen plasma storage [70].

In another article from 2019, Hromadnikova et al. indicated a higher predictive value of C19MC miRNAs expression assay using maternal serum exosomes, compared to assaying C19miRNAs expression in whole maternal serum samples. The selected miRNAs expressed only in the placenta with the highest predictive value (miR-516b-5p, miR-517-5p, miR-518b, miR-520a-5p, miR -525-5p) were analyzed in the samples from patients during the first trimester of pregnancy. In patients who subsequently developed GH or PE, decreased expression of miR-517-5p, miR-520a-5p and miR-525-5p was observed. Moreover, decreased expression of miR-520a-5p was found to be correlated with FGR. An important observation is the convergence of the results from maternal serum exosomes with the level of miRNA expression in the postpartum placenta [39].

Analysis of the expression level of C19MC miRNAs in placental tissues obtained after delivery also appeared in an earlier original study by Hromadnikova et al., from 2015. The expression of 15 miRNAs was assessed when the research team was attempting to determine correlations with the development of GH, PE, FGF. In the work, correlation was found between the decreased expression of miR-517-5p, miR-519d, miR-520a-5p and miR-525 and the development of GH and between the decreased expression of miR-517-5p, miR-518f-5p, miR-519a, miR-519d, miR-520a-5p and miR-525 and the development of FGF. Accordingly, the development of PE was associated with a decrease in the expression of miR-515-5p, miR-517-5p, miR-518b, miR-518f-5p, miR-519a, miR-519d, miR-520a-5p, miR-520h, miR-524-5p, miR-525 and miR-526a and was more pronounced the longer this complication of pregnancy lasted. Downregulation of miR-519a expression was also found to be strongly associated with development of severe pre-eclampsia (sPE) [58].

In contrast, in a study on the analysis of the expression profile of the placental miRNAome in all three trimesters, Inno et al. observed in pregnancies complicated with PE, an increase in the expression of 13 C19MC miRNAs with a negative correlation with gene expression. The strongest correlation was found for the expression of miR-522-5p and miR-518a-5p [63].

A careful analysis of the role of individual C19MC miRNAs and their target genes may explain at what stage and how they are involved in the development of pre-eclampsia. Buckberry et al. tried to systematize the knowledge about the importance of the variable expression of C19MC miRNAs in the development of pre-eclampsia through the analysis of selected target genes [26]. Their studies consistently showed an increase in expression of eight of the C19MC miRNAs during the development of pre-eclampsia. In addition, it was shown that miR-520g and miR-520h inhibited the expression of VEGF and a simultaneous increase in the expression of the VEGF receptor gene, FLT1, in pre-eclamptic placentas [90]. In turn, the importance of increased expression of selected miRNAs C19MC in pre-eclampsia in terms of apoptosis regulation or modification of pro-apoptotic factors is probably related to the *CDKN1A* (*p21*) gene. For miR-519b, 519e, miR-520h and possibly miR-517a, the *CDKN1A* (*p21*) gene is a target gene [91]. *CDKN1A* (*p21*) is associated with apoptosis and plays a role as the cell cycle inhibitor.

The correlation between the increase in the expression of C19MC miRNAs in the course of pre-eclampsia, regardless of the time of symptom onset and their severity, is also probably related to target genes, most of which are genes involved in the processes of immune regulation or inflammatory response in the human body [92]. When comparing the expression level of C19MC and the level of genes regulated by miRNAs, a negative correlation can be seen. Hromadnikova et al. placed PAPPA among the mentioned target genes, the expression of which is regulated by miR-517 *. In their work, the PAPPA protein was used in the first trimester screening test [85].

In further work, upregulated miR-519d was suggested to supposedly silence the expression of *MMP2, CXCL6, NR4A2 and FOXL2*, and was found to be involved in cellular migration and invasion [43,72,93]. In addition, miR-520g is thought to partially inhibit MMP2 synthesis [74], which may lead to impaired remodeling of spiral arteries and thus contribute to the occurrence of PE.

The work of Zhang et al. showed that the target genes for miR-515-5p included hCYP19A1/aromatase, transcription factor glial cells lacking 1 and the WNT receptor ‘frizzled 5’. According to the study, the aforementioned factors played important roles in the process of trophoblast differentiation in early pregnancy [60].

Logan et al. pointed out that the Cajal Bodies marker protein (coilin) was a positive regulator of miR-517-3p biogenesis and was induced by hypoxia. Their study suggested that high expression level of miR-517-3p inhibited the translation of TNFAIP3-Interacting Protein 1 (TNIP1), an inhibitor of the NF-kappa B signaling pathway. Moreover, high levels of miR-517-3p and NF-kappa B inhibit trophoblast invasion and increase sFlt-1 secretion at the same time [64].

Liu et al. showed that miR-518b stimulated trophoblast cell proliferation via the Rap1b-Ras-MAPK pathway. What is more, an increase in the level of observed miR-518b in the PE placenta may lead to excessive trophoblast proliferation [94]. In turn, Canfield et al. investigated the importance of RNA-binding protein LIN28B in the development of PE. They found that there was a decrease in the level of LIN28B in the placenta of women with PE, as compared to normal pregnancies, and that the value of LIN28B was lower the longer the complication lasted [62].

In related work, the knockdown of LIN28B in the JEG3 cell line was seen to reduce cell proliferation, suppress the syncytin 1 (SYN-1) involved in syncytiotrophoblast formation and the apelin receptor endogenous ligand (ELABELA), to decrease C19MC miRNA expression (miR516a, miR-516b and miR-519d) and increase mRNA expression ITGb4 and TNF-a, also influencing the process of inflammation in the placenta [62]. It is believed that the effect of LIN28 on the regulation of C19MC miRNAs expression results from direct binding to the consensus DNA sequence in the promoter regions, and from activation of CpG TET1 demethylase, as well as from binding to the CpG-rich Alu repeats distributed in C19MC, which act as independent promoters of RNA polymerase III [62,95,96].

Liu et al., in other work, showed that miR-520c-3p could block inflammasome activation and the development of the inflammatory cascade in pre-eclampsia by downgrading NLRP3 expression [97]. Xie L. and Sadovsky Y. revealed that miR-519d is a factor that regulated the expression of protein genes involved in the interactions of cells with the extracellular matrix and the processes of migration and thus cell invasion, with no effect on proliferation and apoptosis [61]. Of interest in this work is that it demonstrates the importance of the cell migration process in invasion during trophoblast implantation and infiltration, as well as metastasis during neoplasm [38,74]. The importance of selected C19MC miRNAs in the process of trophoblast development and their modulating effect on the maternal immune response are summarized in Figure 2.

Data from various studies assessing the expression level of C19MC microRNAs in tumors indicate their role in regulating cell proliferation, angiogenesis and possible oncogenic or suppressor activity [98,99,100]. An important common feature of the early stage of placenta development (5–12 hbd) and tumor biology is intense angiogenesis under conditions of relative hypoxia [101,102].

In some aggressive brain tumors, C19MC amplification has been shown [103], while in some thyroid adenomas, the 19q13 region has been involved in chromosomal translocations [30]. Depending on the type of tumor, miR-519d has either an oncogenic or a tumor suppressive function [68]. Here, oncogenic function is related to miR-519d upregulation and has been demonstrated in hepatocellular carcinoma (HCC) [104,105], cervical cancer [100,106] and multiple myeloma [107]. In contrast, tumor suppression function was confirmed in hepatocellular carcinoma [108], lung adenocarcinoma [109], human osteosarcoma [110], ovarian cancer [111], breast cancer [99,112] and chondrosarcoma [113]. Other studies indicate that the regulatory functions of individual C19MC miRNAs in cancer are related to silencing the expression of factors and signaling pathways related to adhesion, migration, differentiation, growth and angiogenesis (Rap1b, ABCG2, DAPK2, ephrins-EphB2 and EphB4, CXCR4) [98,99,100,114,115,116,117].

## 6. Conclusions

The early identification of women at high risk of developing PE is important for monitoring the course of pregnancy, planning perinatal care in a reference center, reducing maternal and fetal mortality and mortality and possibly reducing the likelihood of long-term health effects in the offspring.

The most correct concept seems to be combining biochemical and biophysical tests, among others, the assessment of maternal risk factors, mean blood pressure, uterine Doppler examination and maternal serum biomarkers (related to pregnancy plasma proteins A and placental growth factor [118,119] and placental-specific miRNAs), which would help identify the highest risk group for developing pre-eclampsia, with the lowest possible false-positive rate. There is evidence that the C19MC is involved in the trials important for the proper placement and development of the placenta, and their dysregulation results in disturbed differentiation, trophoblast invasion and angiogenesis. Analyses of PubMed articles on the role of C19MC miRNAs in the pathogenesis of pre-eclampsia indicate a significant increase in the expression of most C19MC miRNAs in placental tissue and a high level of circulating fractions in serum and plasma [34,38,66,67,68,69,70,72,73,74,75,76,77,78,79,80]. An increase in C19MC RNA expression was demonstrated in both the first and/or third trimester, noting a sharp decrease in the plasma or serum levels of miRNA in postpartum women [34,66]. Only for miR-525-5p, were low levels of plasma expression noted in the first trimester, and in the placenta in the third trimester [39,77].

Knowledge about the target genes of miRNAs belonging to C19MC offers another opportunity to diagnose patients at risk of PE before the onset of clinical symptoms, and, in the future, to develop new PE therapies [35].

## Figures and Tables

**Figure 1 ijms-23-13836-f001:**
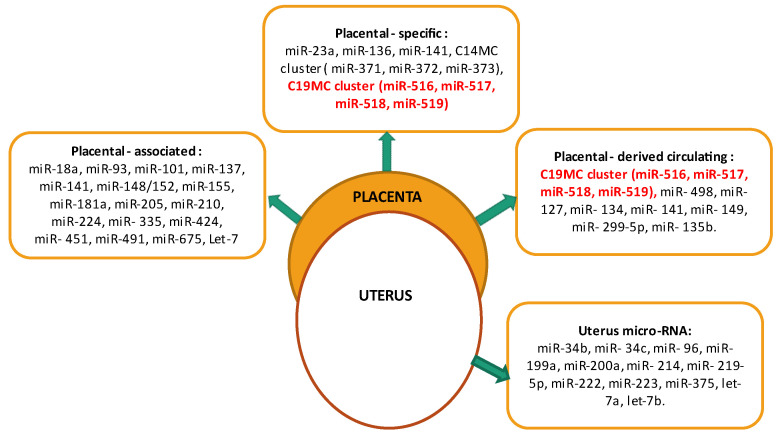
Division of miRNAs correlated with PIRC: placental-associated, placental-specific, placental-derived circulating and uterus micro-RNA [13,14,15].

**Figure 2 ijms-23-13836-f002:**
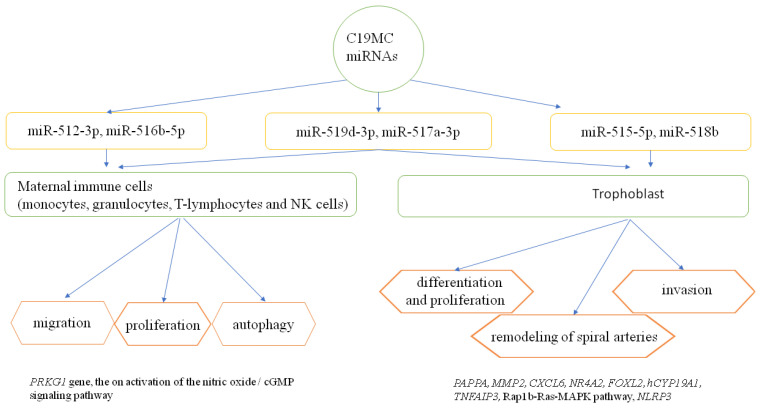
The role of selected C19MC miRNAs in the regulation of trophoblast development and the maternal immune response [38,54,60,63,67,73,83,84,85,87]. *PRKG1*—protein kinase, cGMP- dependent, regulatory, type 1; *PAPPA*—pappalysin 1, pregnancy-associated plasma protein A; *MMP2*—matrix metalloproteinase 2; *CXCL6*—chemokine, CXC motif, ligand 6; *NR4A2*—nuclear receptor subfamily 4, group A, member 2; *FOXL2*—forkhead transcription factor FOXL2; *hCYP19A1*—human cytochrome P450, family 19, subfamily A, polypeptide 1; *TNFAIP3*—tumor necrosis factor-alpha-induced protein 3; *Rap1b-Ras-MAPK* pathway-RAS-related protein Rap1b-Ras-mitogen-activated protein kinases pathway; *NLRP3*—NLR family, pyrin domain-containing 3.

**Table 1 ijms-23-13836-t001:** The list of analyzed articles published in 2013–2022 in PubMed, searched on the basis of keywords: C19MC and pre-eclampsia [26,28,35,38,39,43,56,57,58,59,60,61,62,63,64].

Authors:	Type of Article:	Materials and Methods:	miRNAs:	Conclusions:
Hromadnikova et al., (2013)	Research article	Ultrasound examination with Doppler study Maternal plasma samples Placental tissue Quantitative real-time PCR	miR-516-5p miR-517 * miR-518b miR-520a * miR-520h miR-525 miR-526a	There is no relationship between plasma concentration of C19MC miRNA in pregnant women and abnormal perinatal results. In complicated pregnancies, an inverse relationship was found between the expression level of miR-516-5p, miR-517 *, miR-520a *, miR-525 and miR-526a and the flow rate in the central artery of the brain. The increase in expression of miR-516-5p, miR-517 *, miR-520a *, miR-525, and miR-526a is characteristic of pre-eclampsia. The circulating C19MC miRNAsprobably do not play a significant role in the pathogenesis of GH and FGR.
Buckberry et al., (2014)	Review article	Analysis of the results of the latest genome scale and single gene expression research using mouse or human placental tissues		ncRNAs, including those undergoing epigenetic regulation, in the genomic imprinting mechanism, and XCl are involved in the development of pre-eclampsia. An important question, however, is whether the change in miRNA expression levels is a cause or a consequence of PE.
Hahn et al., (2014)	Review article	Analysis of published data		Hypomethylated cell-free placenta-derived DNA is likely to activate maternal effector cells of the immune system to produce inflammatory cytokines (IL-6) via TLR-9. The high level of cell-free DNA levels in pregnancy may increase the likelihood of thromboembolic complications. Primate placenta-specific C19MC miRNAs may affect the maternal immune response in the course of viral infection.
Hromadnikova et al., (2015)	Original research article (retrospective study)	Doppler ultrasonography results Placental tissues Quantitative real-time PCR	miR-512-5p miR-515-5p miR-516-5p miR-517-5p miR-518b miR-518f-5p miR-519a miR-519d miR-519e-5p miR-520a-5p miR-520h miR-524-5p miR-525 miR-526a miR-526b	Correlation was found between the decreased expression of miR-517-5p, miR-519d, miR-520a-5p and miR-525, and the development of GH. Correlation was found between the decreased expression of miR-517-5p, miR-518f-5p, miR-519a, miR-519d, miR-520a-5p and miR-525, and the development of FGF. Correlation was found between the decreased expression of miR-515-5p, miR-517-5p, miR-518b, miR-518f-5p, miR-519a, miR-519d, miR-520a-5p, miR-520h, miR-524-5p, miR-525 and miR-526a, and the development of PE. Downregulation of miR-519a expression is associated with the development of severe pre-eclampsia. Correlation was found between the duration of the pregnancy-related complication and the downregulation of miR-515-5p, miR-518b, miR-518f-5p, miR-519d and miR-520h.
Miura et al., (2015)	Research article	Maternal plasma samples at 27–34 weeks of gestation Quantitative real-time PCR	miR-518b miR-516b miR-516a-5p miR-525-5p miR-515-5p miR-520h miR-520a-5p miR-519d and -526b	The levels of miR-518b, -1323, -516b, -516a-5p, -525-5p, -515-5p, -520h, -520a-5p, and -526b are higher in the plasmas of pregnant women with sPELO. The levels of miR-1323, -516b, -516a-5p, -525-5p, -515-5p, -520h, -520a-5p, and -526b are higher in the plasmas of pregnant women with sPEEO. The increase in the level of C19MC miRNAs in the plasmas of pregnant women with sPE is probably a consequence of the development of this complication.
Zhang et al., (2016)	Research article	Primary human CytT from mid-gestation placenta Quantitative real-time PCR Immunoblot	miR-515-5p miR-519e-5p miR-519c -3p miR-518f	miR-515-5p is markedly decreased during differentiation of human syncytiotrophoblasts and significantly increased in the placenta of women suffering from pre-eclampsia. Overexpression of miR-515-5p inhibits syncytiotrophoblast differentiation. Target genes for miR-515-5p include hCYP19A1/aromatase, transcription factor glial cells lacking 1, and the WNT receptor, frizzled 5.
Xie L, Sadowsky Y (2016)	Review article	Analysis of published data	miR-519d	miR-519d is involved in the regulation of biosynthesis proteins involved in cell–extracellular matrix interaction and thus also influences the migration and invasion of cells. miR-519d is involved in the regulation of migration and possible EVT invasion during PE development.
Hromadnikova et al., (2017)	Research article	Maternal plasma samples collected in the 1st trimester (10–13 hbd) Quantitative real-time PCR	miR-516b-5p miR-517-5p miR-518b miR-520a-5p miR-520h miR-525-5p	Increased expression of miR-517-5p, miR-518b and miR-520h in the plasma of pregnant women tested in the first trimester (11-13 hbd) who developed pre-eclampsia, was observed. miR-517-5p has the highest predictive value. Between the level of C19MC expression and the risk of IUGR, there is no correlation.
Malnou et al., (2019)	Review article	Analysis of published data		C19MC miRNAs packed in exosomes influence even distant cells, affecting gene translation. Recent research shows the importance of epigenetically regulated miRNAs gene clusters in the placenta, as well as cell-to-cell communication, and development of pregnancy complications.
Hromadnikova et al., (2019)	Research article	Exosomes isolated from maternal plasma samples collected in the 1st trimester (10–13 hbd) Real-time PCR	miR-516b-5p miR-517-5p miR-518b miR-520a-5p miR-525-5p	Decreased expression of miR-517-5p, miR-520a-5p, miR-525-5p in patients who subsequently developed GH or PE. Decreased expression of miR-520a-5p is correlated with FGR.
Canfield et al., (2019)	Research article	Placental tissue in the 1st trimester JEG3 cell line Immunohistochemistry Immunoblotting Quantitative real-time PCR	miRNAs C19MC (among others, miR516a, miR-516b, miR-519d)	LIN28B mRNA and protein expression levels are clearly lowered in pre-eclampsia. Knockdown LIN28B in the JEG3 cell line reduced cell proliferation, suppressed syncitin 1 (SYN-1) and the apelin receptor endogenous ligand (ELABELA) and decreased expression of C19MC miRNAs (miR516a, miR-516b, and miR-519d) and increased mRNA expression. ITGb4 and TNF-a. LIN28B is possibly the dominant paralog in the human placenta. The decreased level of LIN28B plays a significant role in the development of PE through inhibition of trophoblast invasion and syncytialization, and onset of inflammation.
Morales-Prieto et al., (2020)	Review article	Analysis of articles published in PubMed in 2010–2020. Search scope: ”microRNA” and ”placenta”		The development and function of the human placenta is regulated through an intricate network of over 1900 miRNAs. miRNAs primarily affect the metabolism, proliferation, differentiation and invasion of trophoblast cells. The transfer of miRNAs packed into EVs is fundamental to the fetal-mother dialogue during pregnancy.
Munjas et al., (2021)	Review article	Analysis of published data	C19MC lncRNA lnRNA	C19MC dysregulation leads to dysfunction of trophoblast cells, abnormal placentation and the consequent development of PE. C19MC miRNAs are involved in maternal regulation of the immune system during pregnancy. Trophoblast-derived exosomes containing miRNAs may affect the immune system of pregnant women, the development of a systemic inflammatory response and possible progression of PE.
Inno et al., (2021)	Original research	52 placentas of 3 trimesters Small-RNA sequencing	417 miRNAs (including miRNAs of C19MC)	miR-143-3p is a potential marker for placental maturation or pregnancy progression. Two thirds of the C19MC miRNA is expressed especially in the first trimester, is very low in the second trimester and slowly increased in the third trimester. Most of the C19MC miRNA target genes are involved in the cell signaling or transcription regulation important in early pregnancy. In PE, an increase in 13 C19MC miRNAs was observed with a negative correlation to gene expression. The highest correlation was found for miR-522-5p and miR-518a-5p.
Logan et al., (2022)	Research article	Animal models (rats at gestation day 11) JEG-3 cell line Western blotting Quantitative real-time PCR Immunohistochemistry	miR-517-3p	Coilin (the Cajal Bodies marker protein) is a positive regulator of miR-517-3p biogenesis, induced by hypoxia. High expression level of miR-517-3p inhibits the translation of TNFAIP3 Interacting Protein 1 (TNIP1), an inhibitor of the NF-kappa B signaling pathway. High levels of miR-517-3p and NF-kappa B inhibit trophoblast invasion and increase sFlt-1 secretion at the same time.

**Table 2 ijms-23-13836-t002:** Summarizes the data from the analyzed articles assessing the expression of individual C19MC miRNAs in pre-eclampsia by source (plasma, serum, placenta), sampling time and expression level [34,38,39,56,58,60,63,66,67,68,69,70,71,72,73,74,75,76,77,78,79,80].

No	Author	Sample	Sampling Time	Analyzed miRNAs C19MC	Expression Level in PE
1.	Luo (2009)	Placenta Maternal plasma	1st trimester (7–11 w) 3rd trimester (36-38 w) 1 day before delivery 3 days after delivery	miR-517a, miR-517b, miR-518b, miR-519a, miR-512-3p	↑ in 1st trimester in placentas; in maternal plasma markedly decreased after delivery.
2.	Miura (2010)	Placenta Plasma Maternal plasma	1st trimester (12–13 w) 3rd trimester (38–39 w) 1 day after delivery	miR-515-3p miR-517a miR-517c miR-518b miR-526b	↑ in placenta, higher than in plasma; significantly decreased after delivery
3.	Morales-Prieto (2012)	Placenta	1st trimester 3rd trimester (the day of delivery)	miR-512 (-3p,-5p) miR-515 (-3p,-5p) miR-516 (a-5p, b) miR-517 (a *, b, c) miR-518 (b, c, c *, d-5p, e, e *, f, f *) miR-519 (a, b- 3p, c-3p, d, e, e *) miR-520 (a-3p, a-5p, b, c-3p, d-3p, d-5p, e, f, g, h) miR-521 miR-522 miR-523 miR-524 (-5p) miR-525 (-3p, -5p) miR-526 (b)	↑ from 1st to 3rd trimester
4.	Flor (2012)	Placenta	1st trimester (10–14 w)	miR-520-3p miR-519a-3p miR-517a-3p	↑ (highly expressed in 1st trimester placentas)
5.	Wang (2012)	Placenta	1st trimester (6–9 w)	miR-517b miR-519a	↑
6.	Hromadnikova (2012)	Plasma	Retrospective study: Samples collected at various gestational stages	miR-516-5p miR-517 * miR-518b miR-520a * miR-520h miR-525 and miR-526a	No correlation
7.	Hromadnikova (2013)	Plasma Placenta	3rd trimester (the day of delivery: before and after 34 hbd)	miR-516-5p miR-517 * miR-518b miR-520a * miR-520h miR-525 miR-526a	↑ ↑ ↑ ↑ ↑
8.	Ura (2014)	Serum	1st trimester (12–14 w)	miR-520a	↑
9.	Xie (2014)	Placenta	1st trimester (6–12 w)	miR-517-3p miR-518b miR-519d miR-520g miR-515-5p	↑
10.	Yang S (2015)	Placenta Plasma	The day of delivery (by elective caesarean section)	miR-517c miR-518-3p miR-518e miR-519d	↑ in the plasma and placenta
11.	Hromadnikova (2015)	Placenta	Retrospective study: Samples collected on the day of delivery	miR-512-5p miR-515-5p miR-516-5p miR-517-5p miR-518b miR-518f-5p miR-519a miR-519d miR-519e-5p miR-520a-5p miR-520h miR-524-5p miR-525 miR-526a miR-526b	↓ ↓ ↓ ↓ ↓ ↓ ↓ ↓ ↓ ↓ ↓
12.	Zhang (2016)	Placenta	The day of delivery	miR-525	↑
13.	Hromadnikova (2017)	Plasma	1st trimester (10–13 w)	miR-517-5p miR-518b miR-520h	↑
14.	Jiang L (2017)	Serum	1st trimester (8–10 w)	miR-520g	↑
15.	Takahashi (2017)	Placenta	1st trimester (7–11 w)	miR-520c-3p	↑ in the maternal decidua stroma;
16.	Fu JY (2018)	Placenta	3rd trimester (the day of delivery)	miR-517-5p	↑
17.	Hromadnikova (2019)	Exosomes from Plasma	1st trimester (10–13 w)	miR-520a-5p miR-525-5p miR-517-5p	↓ ↓ ↓
18.	Zhang M (2020)	Placenta	3rd trimester (the day of delivery)	miR-525-5p	↓
19.	Mong (2020)	Placenta	1st trimester (7–8 w)	miR-517a miR-517c	↑
20.	Gonzalez (2021)	Placenta	1st trimester 3rd trimester (the day of delivery)	18 from C19MC: miR-512-5p miR-515-5p miR-516a-3p miR-516a-5p miR-516b-3p miR-517b-3p miR-517-5p miR-518b miR-518c-3p miR-518f-3p miR-520c-3p miR-519c-5p 52 miR-520f-3p miR-520e miR-520d-5p miR-520g-5p miR-520d-3p miR-520c miR-521 miR-524-3p miR-526b-3p miR-527	↑ from 1st to 3rd trimester
21.	Inno (2021)	Placenta	1st trimester 2nd trimester 3rd trimester (the day of delivery)	417 tested miRNAs Including miRNAs C19MC: miR-522-5p miR-528-5p	↑ ↑ ↑
22.	Nunode (2022)	Plasma Placenta	3rd trimester (the day of delivery)	miR-515-5p	↑

## Data Availability

Detailed data available via email: ilona.krupa@interia.pl.

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
