# Peer review of "The Role of Cluster C19MC in Pre-Eclampsia Development"

_ijms, 2022, doi:10.3390/ijms232213836_

Round 1

Reviewer 1 Report

In the present review, the Authors disclosed current knowledge on C19MC miRNAs role in preeclampsia development. This is an interesting review addressing an important issue as C19MC involvement for a proper placement and development of the placenta, and as their dysregulation resulted in disturbed differentiation, trophoblast invasion and angiogenesis. Therefore, it is of great interest to the readers of IJMS.

However, there are several issues that need to be addressed before publication.

1.      Please, update Preeclampsia definition.

2.      Table 1 is part of the Results section that the Authors completely missed.

3.      It could be very useful to introduce some paragraphs to classify  C19MC miRNA and results obtained such as “Placental miRNA”, “Circulating (serum) miRNA”….

4.      The Authors mentioned that C19MC miRNA are directly involved in placental invasion, angiogenesis and immune response regulation. It is very interesting since PE is characterized by the alteration of all these molecular processes. You should explain better from a molecular point of view how C19MC miRNA modulate angiogenesis (e.g. sFlt-1, PLGF, VEGF), invasion and immune response.

5.      The Authors mentioned that  C19MC miRNAs are epigenetically regulated. This point is very important and you should add more details. Moreover, It is well know that epigenetic modification are reversible. Do you have any comment on this?

6.      There are differences between placental and maternal PE in C19MC miRNAs expression?

7.      It could be very useful to add a table with an overview of C19MC miRNA in Preeclampsia indicating the source (plasma, placenta…) and if it is up or down-regulated.

Author Response

Reviewer #1

However, there are several issues that need to be addressed before publication.

  1. Please, update Preeclampsia definition.

 The definition of Preeclampsia has been corrected and updated.

  1. Table 1 is part of the Results section that the Authors completely missed.
    The layout of the manuscript was corrected and supplemented by adding the section: Results.
  2. It could be very useful to introduce some paragraphs to classify  C19MC miRNA and results obtained such as “Placental miRNA”, “Circulating (serum) miRNA”….
    As suggested, the manuscript was supplemented with information on the division of miRNAs.
  3. The Authors mentioned that C19MC miRNA are directly involved in placental invasion, angiogenesis and immune response regulation. It is very interesting since PE is characterized by the alteration of all these molecular processes. You should explain better from a molecular point of view how C19MC miRNA modulate angiogenesis (e.g. sFlt-1, PLGF, VEGF), invasion and immune response. – The general information has been added in the section: Introduction.

And the effect of increased expression of selected C19MC miRNAs on the modulation of angiogenesis, apoptosis and changes in the immune response during the development of preeclampsia were presented after completing the manuscript in the section: Discussion and Figure 2.

  1. The Authors mentioned that  C19MC miRNAs are epigenetically regulated. This point is very important and you should add more details. Moreover, It is well know that epigenetic modification are reversible. Do you have any comment on this?
    As suggested, information on C19MC epigenetic regulation has been added .
  2. There are differences between placental and maternal PE in C19MC miRNAs expression?
    The manuscript has been supplemented with data in the Results and Conclusions sections.
  3. 7.It could be very useful to add a table with an overview of C19MC miRNA in Preeclampsia indicating the source (plasma, placenta…) and if it is up or down-regulated.

As suggested, the table summarizes the data from the analyzed articles assessing the expression of individual C19MC miRNAs in pre-eclampsia by source (plasma, serum, placenta), sampling time, and expression level was added.

Reviewer 2 Report

The article «The role of cluster C19MC in pre-eclampsia development» summarizes current data on the role of C19MC miRNAs in the development of pre-eclampsia. The authors summarized some articles by keywords («C19MC and pre-eclampsia») in Pubmed. However, this is not enough since it is necessary to search for each microRNA separately.

The authors might improve the Manuscript:

Major:

1.      Table 1 is not illustrative. The section «Conclusions» needs to be redone.

2.      The article lacks a generalizing analysis of the results. It is necessary to expand the range of analyzed articles, to add new information to the chapter «results» and «conclusion».

A) How are individual microRNAs of the C19MC cluster expressed during all trimesters of pregnancy?

B) In which biomaterial does microRNA expression occur (blood, urine of pregnant women)? If there is blood, then plasma or serum is used for detection? Are the authors aware that the detection of the same microRNAs in plasma and serum may show different results?

C) At what time, 1st, 2nd or 3rd trimester (or in the placenta) can we, based on different data, talk about the general patterns of expression of all microRNAs in this cluster?

Minor:

1.      Line 2-3 The word «review» should be deleted.

2.      In Abstract: You should expand and rewrite the results

3.      Line 74 What means “C19MC”? Is that the title of the chapter?

4.      Line 123-125 You write that «The scope of the search was determined on the basis of the following keywords: ‘C19MC and pre-eclampsia’, obtaining the result of 16 articles, and ‘C19MC and pre-eclampsia’, obtaining the result of 11 articles» What are the differences?

5.      Line 331 «Figures and Tables with Captions» should be deleted.

Author Response

  1. Table 1 is not illustrative. The section «Conclusions» needs to be redone.

The layout of the manuscript was corrected and supplemented by adding the section: Results. We prepared a new Table 2 and corrected Table 1.

  1. The article lacks a generalizing analysis of the results. It is necessary to expand the range of analyzed articles, to add new information to the chapter «results» and «conclusion».

We added a new information in results section and conslusion.

  1. A) How are individual microRNAs of the C19MC cluster expressed during all trimesters of pregnancy?
  2. B) In which biomaterial does microRNA expression occur (blood, urine of pregnant women)? If there is blood, then plasma or serum is used for detection? Are the authors aware that the detection of the same microRNAs in plasma and serum may show different results?
  3. C) At what time, 1st, 2nd or 3rd trimester (or in the placenta) can we, based on different data, talk about the general patterns of expression of all microRNAs in this cluster?

As suggested, the scope of the analyzed articles was extended based on the Pubmed search using individual C19MC miRNAs and "preeclampsia" as keywords. Summary data on the expression of individual C19MC miRNAs in pre-eclampsia by source (plasma, serum, placenta), time of collection samples and expression levels are included in Table 2.

The section: Conlusions has been supplemented.

Minor:

  1. Line 2-3 The word «review» should be deleted- done

The manuscript has been corrected.

  1. In Abstract: You should expand and rewrite the results. We rewrite results
  2. Line 74 What means “C19MC”? Is that the title of the chapter?
    Yes, that is the title of the chapter. I corrected the spelling and marked it as the 2nd chapter in the text.
  3. Line 123-125 You write that «The scope of the search was determined on the basis of the following keywords: ‘C19MC and pre-eclampsia’, obtaining the result of 16 articles, and ‘C19MC and pre-eclampsia’, obtaining the result of 11 articles» What are the differences?

The difference is in the spelling of "pre-eclampsia" vs. "preeclampsia", both forms of writing appear in publications. However, including these forms as keywords resulted in slightly different search results.

5.      Line 331 «Figures and Tables with Captions» should be deleted.
The manuscript was corrected.

Round 2

Reviewer 1 Report

1.      Abstract. In the background section it is not clear the link between PE and C19MC miRNA that is not even mentioned. The Authors should briefly explain that C19MC miRNA is involved in PE pathogenesis. In the Method section of the abstract, the Authors should described better the article they included (e.g. from 2012 to 2021…)

2.      Methods section. Please, add more details on methods you used. When did you perform the research? Which studies did you select? Which is the temporal range you included? Did you exclude some paper?

3.      Table 2. I’m a bit surprised that a lot of study investigated miRNA expression in placental tissues during the first trimester since from an ethical point of view is not approved all over the word (it means that the pregnancy is stopped) or do you consider “placental miRNA” miRNA released by the placenta?  Please check and clarify.

Author Response

Reviewer 1

  1. In the background section it is not clear the link between PE and C19MC miRNA that is not even mentioned. The Authors should briefly explain that C19MC miRNA is involved in PE pathogenesis. In the Method section of the abstract, the Authors should described better the article they included (e.g. from 2012 to 2021…)

The abstract summary has been corrected and completed as suggested.

  1. Methods section. Please, add more details on methods you used. When did you perform the research? Which studies did you select? Which is the temporal range you included? Did you exclude some paper?

A search of the Pubmed database for Table 1 creating was conducted in September 2022.

In turn, the search of the Pubmed database of Table 2 creating was carried out in November 2022.  Using the above-mentioned keywords, a total of 30 records were obtained. Some articles have been duplicated for different miRNAs. Taking this into account, 17 articles were ultimately analyzed. 4 articles were excluded: 1 written in Czech (Hromadnikova et al., 2010), 2 experimental works on cell lines (Logan et al., 2022; Shi et al., 2022), 1 review (Sheikh et al., 2016 ). The remaining 10 articles listed in Table 2 were included based on a systemic review by Ogoyama et al. (2022) [57, 34, 81, 58, 59, 60, 99, 62, 63, 65, 66, 67, 68].

  1. Table 2. I’m a bit surprised that a lot of study investigated miRNA expression in placental tissues during the first trimester since from an ethical point of view is not approved all over the word (it means that the pregnancy is stopped) or do you consider “placental miRNA” miRNA released by the placenta? Please check and clarify.

The term "placental miRNA" means miRNAs, including C19MC miRNAs, that are expressed and can be isolated from placental tissue.

Table 2 presents 10 experimental studies analyzing placental miRNAs in placental tissue collected in the first trimester. The examined 1st trimester placental tissues were collected by: CVS procedures (Flor 2012, Gonzalez 2021), during legal termination of pregnancy (Luo 2009, Miura 2010, Wang 2012, Takahashi 2017, Unno 2021). Archival material was used in 3 analyzes, but no detailed data are available (Morales - Prieto 2012, Xie 2016, Mong 2020).

Detailed informations from individual articles are provided below:

From Luo (2009): „Human placentas and blood plasma samples were obtained according to protocols approved by the Nippon Medical School Hospital Ethics Committee and the Jichi Medical University Ethics Committee. First-trimester and full-term placental tissues were obtained from elective terminations of pregnancy and uncomplicated cesarean deliveries, respectively.”

From Miura (2010): „For the miRNA microarray analysis, we used 2 sets of placental tissue samples and corresponding maternal blood samples collected in the first trimester (12–13 weeks of pregnancy) and another 2 sets collected in the third trimester (38–39 weeks). Placental tissue samples were obtained immediately after the termination of pregnancy and placed in RNAlater™ (Ambion/Applied Biosystems). Blood samples (7 mL) were collected before the termination of pregnancy and placed in tubes containing EDTA. miRNA was extracted from placental tissues and maternal blood cells immediately after sampling.”

From Morales-Prieto (2012): „ Total RNA was obtained from isolated cytotrophoblast cells from healthy term and first trimester placentae and the cell lines HTR-8/SVneo (immortalized trophoblast cells), JEG-3 (choriocarcinoma), ACH-3P and AC1-M59, which are choriocarcinoma cells fused with first and third trimester trophoblast cells, respectively.”

From Flor (2012): ” For cell cultures of chorionic villi, first-trimester samples (10–14 weeks of gestation) were collected by transabdominal chorionic villus sampling (CVS) for prenatal cytogenetic diagnosis. Cell cultures were set up following routine methods for establishing cultures of chorionic villi stromal cells as, e.g., outlined by Portmann-Lanz et al. [18] and Yong et al. [19] with a few modifications. A cytogenetic analysis was performed by standard procedures. The surplus cell cultures were donated to this study after.”

From Wang (2012): „ The use of placental tissues was approved by the local ethical committee. Placental tissues from 100 cases each at 6, 7, 8 or 9 gestational weeks were collected following elective terminations at Shengjing Hospital of China Medical University informed written consent given by the patients.”

From Xie (2016): „The Institutional Review Board at the University of Pittsburgh approved the collection and analysis of de-identified specimens under an exempt protocol. We used formalin-fixed, paraffin-embedded archival placental samples from the first trimester (6 –12 wk), term (37– 41 wk) pregnancy, and ectopic (tubal) pregnancy. (…) First trimester cytotrophoblasts were derived from a pool of proliferating trophoblasts prepared from seven first-trimester placentas (8 –10 wk).”

From Takahashi (2017): ” The study was performed according to the guidelines of the Declaration of Helsinki. Human placentas from patients who gave informed consent were obtained using protocols approved by the Nippon Medical School Ethics Committee and the Jichi Medical University Ethics Committee. First trimester placental tissues (at 7e13 weeks of gestation, n ¼ 18) were aseptically obtained after legal abortions.”

From Mong (2020): ” Placental sections from the frst trimester (7- and 8-week gestation, n=2) and early human pregnancies (20-weeks week gestation, n=2) were obtained from a previously banked deidentifed parafn tissues, under approval by Yale University Human Investigation Committee and by the institutional review board of the University of South Florida.”

From Gonzalez (2021): ” Samples from the first trimester of pregnancy were collected between 70–102 days gestation during CVS procedures done for prenatal diagnosis.” 

From Inno (2021): „First and second trimester placental samples had been collected from women who underwent elective surgical termination of pregnancy or medically induced abortion due to maternal medical risks.”

Reviewer 2 Report

The article can be published. The authors corrected the comments

Author Response

The article can be published. The authors corrected the comments.